# Evidence from Buhais Rockshelter for human settlement in Arabia between 60,000 and 16,000 years ago

K. Bretzke [1,2] ✉, S. Kim [3,4], S. A. Jasim[5], E. Yousif[5], F. Preusser[6], G. W. Preston[7], F. Pallottino[7] & A. G. Parker [7] ✉

Several significant milestones in human evolution date to the period between 70,000 and 12,000 years ago, including the replacement of archaic humans, the global dispersal of *Homo sapiens* and the introduction of Upper Palaeolithic traditions. The Arabian Peninsula provides only sparse records illuminating this period. We introduce here the Buhais Rockshelter archaeological sequence and paleoenvironmental records from the Faya Palaeolandscape in the Emirate of Sharjah (UAE). Buhais Rockshelter provides stratified stone artifact assemblages reflecting habitation phases around 125,000, 59,000, 35,000 and 16,000 years ago. Palaeoenvironmental fieldwork further shows that settlement at Buhais Rockshelter is contemporaneous with increased water availability in the landscape at these times. Our results contradict the prevailing view of human absence in Arabia at the end of the Pleistocene and call for reassessing the inhabitability of southern Arabia during the last glacial period. Results from Buhais Rockshelter extend known records from Jebel Faya and demonstrate repeated occupation of the region between 210,000 and 16,000 years ago. Together, this contributes data for a critical timeframe in human evolution providing an empirical foundation for testing anthropological models about human adaptation to and dispersal through the desert landscapes of southern Arabia.

Recent evidence from the Arabian Peninsula has substantially advanced understanding of the region's early prehistory[1–6]. Archaeological records documenting human occupation during wet phases of Marine Isotope Stage (MIS) 5 are particularly well represented (Fig. 1). In contrast, sites post-dating MIS 5 are exceedingly rare. As a result, the timing and nature of human presence on the Arabian Peninsula during the terminal Pleistocene, between approximately 70 ka and 12 ka ago, remain poorly understood. Only two chronometrically dated sites are currently known from the youngest phase of the Late Middle

Palaeolithic (LMP). Wadi Surdud in Yemen and al-Marrat 3 in northern Saudi Arabia provide evidence for human occupation of the Peninsula around 60–50 ka ago[3,7]. This period is of particular importance for human evolutionary history, as genetic evidence suggests that the final successful expansion of modern humans into Southwest (SW) Asia occurred at roughly the same time[8].

In addition to questions surrounding global human dispersal, another major evolutionary milestone remains poorly represented in the archaeological record of the Arabian Peninsula: the emergence of

[1]Seminar for Prehistoric and Protohistoric Archaeology, University of Jena, Jena, Germany. [2]Department of Early Prehistory and Quaternary Ecology, University of Tübingen, Tübingen, Germany. [3]Université Paris 1 Panthéon-Sorbonne, École doctorale d'archéologie (ED 112), Paris, France. [4]UMR 5133, 'Laboratoire Archéorient', Maison de l'Orient et de la Méditerranée, Lyon, France. [5]Sharjah Archaeology Authority, Government of Sharjah, Sharjah, United Arab Emirates. [6]Institute of Earth and Environmental Sciences, University of Freiburg, Freiburg, Germany. [7]Human Origins and Palaeoenvironments Research Group, Oxford Brookes University, Oxford, UK. ✉e-mail: knut.bretzke@uni-jena.de; agparker@brookes.ac.uk

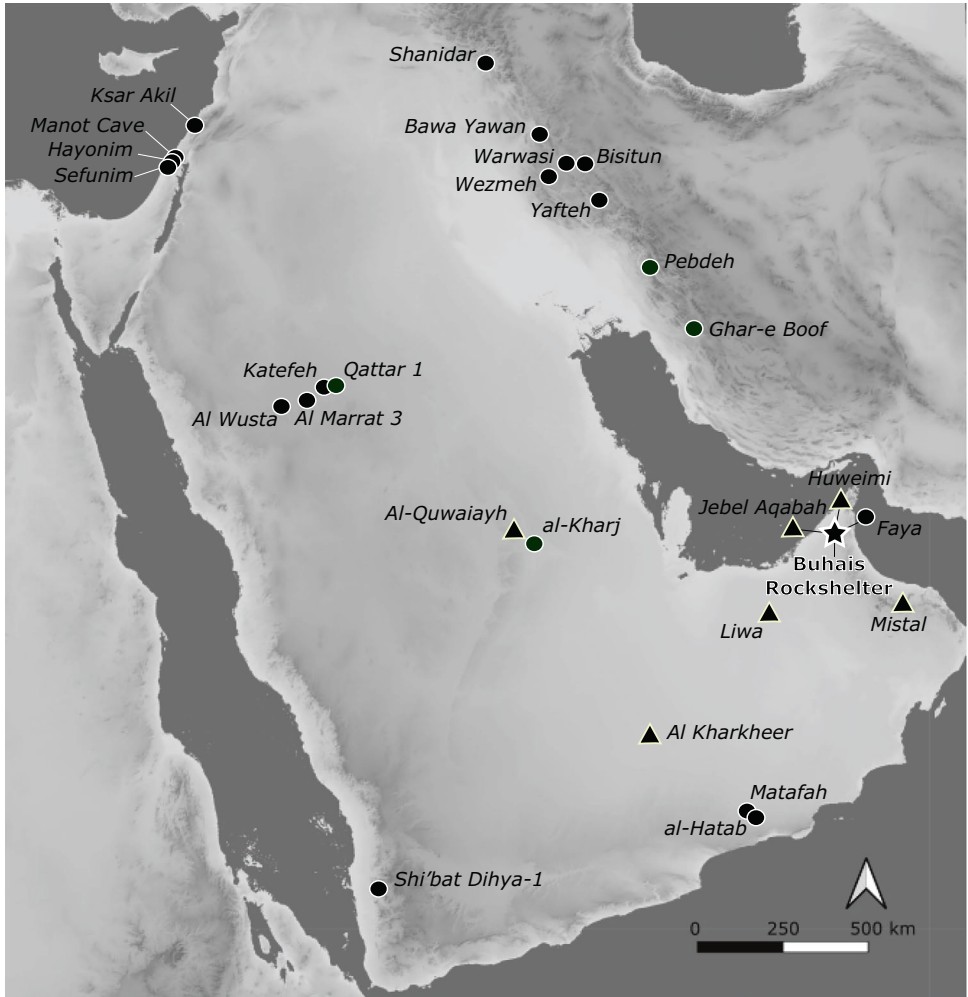

**Fig. 1 | Map of the Arabian Peninsula.** Locations of Buhais Rockshelter (star) as well as key archaeological (circles) and palaeoenvironmental (triangle) sites are shown. Base map developed from public domain data GMTED2010 (courtesy of the U.S. Geological Survey) using QGIS 3.40.

Upper Palaeolithic (UP) traditions. In the Southwest Asia, early UP industries are dated to approximately 50 ka and 35 ka[9–11]. To date Matafah in southern Oman (Dhofar) is the only site in the region that provides both chronometric dates and a lithic assemblage typologically attributed to the UP[12]. No site on the Arabian Peninsula currently preserves stratified evidence spanning both LMP and UP periods, limiting insights into cultural and demographic dynamics during this critical phase of human evolution.

Here, we present evidence from Buhais Rockshelter (United Arab Emirates, UAE), part of the UNESCO World Heritage Site Faya Palaeolandscape, together with data from adjacent areas, contributing to a more detailed understanding of terminal Pleistocene settlement in Southeast (SE) Arabia. These findings provide insights into local palaeoenvironmental conditions and regional habitability between approximately 60–16 ka. Our archaeological records document human occupation just before and after the Last Glacial Maximum (ca. 20 ka), as well as the presence of a fully developed early UP industry in SE Arabia by ca. 35 ka. Combined with palaeoenvironmental reconstructions, these data indicate that SE Arabia remained habitable during at least parts of the terminal Pleistocene and necessitate a reassessment of the region's role in models of human dispersals and adaptation to arid environments during periods traditionally viewed as hyper-arid. Our results further suggest regionally distinct settlement dynamics and changing patterns of spatial connectivity among hunter-gatherer groups occupying SE Arabia during the LMP and UP periods.

## Results

Buhais Rockshelter is situated in the central region of the Emirate of Sharjah (UAE), approximately 60 km inland from the Gulf coast and about 15 km south of the palaeolithic site FAY-NE1 (Fig. 2) at Jebel Faya[13–15]. In contrast to FAY-NE1, Buhais Rockshelter lies on the western side of the Faya-Buhais anticline and facies the extensive sand dunes of the northern Rub' al-Khali. The site is located about 800 m to the east of the main local drainage, Wadi Iddayyah (Fig. 2).

The modern shelter measures approximately 20 m in width and between 1.5 m and 2 m in depth (Fig. 2). It is formed within the Late Cretaceous to Palaeogene Muthaymima Limestone Formation at the southern end of Jebel Buhais. The site is primarily known for Iron Age burial remains, excavated during the 1990s[16]. In contrast, its Pleistocene deposits remained unexplored until the present project, which commenced with test excavations in 2017. Two 1 × 1 m test trenches revealed stratified lithic material, prompting a full-scale excavation beginning in 2019. Over four seasons between 2019 and 2024, an area of 24 m² was excavated to a maximum depth of 1.70 m (Figs. 3, 4).

### Site stratigraphy

Excavation revealed three geological horizons (GH) identified based on their composition and designated GH 1 to 3. GH 1 was further subdivided into four sublayers (a-d), each exhibiting minor sedimentological differences (Fig. 4, Supplementary Fig. 1). All layers are inclined in accordance with the morphology of the underlying

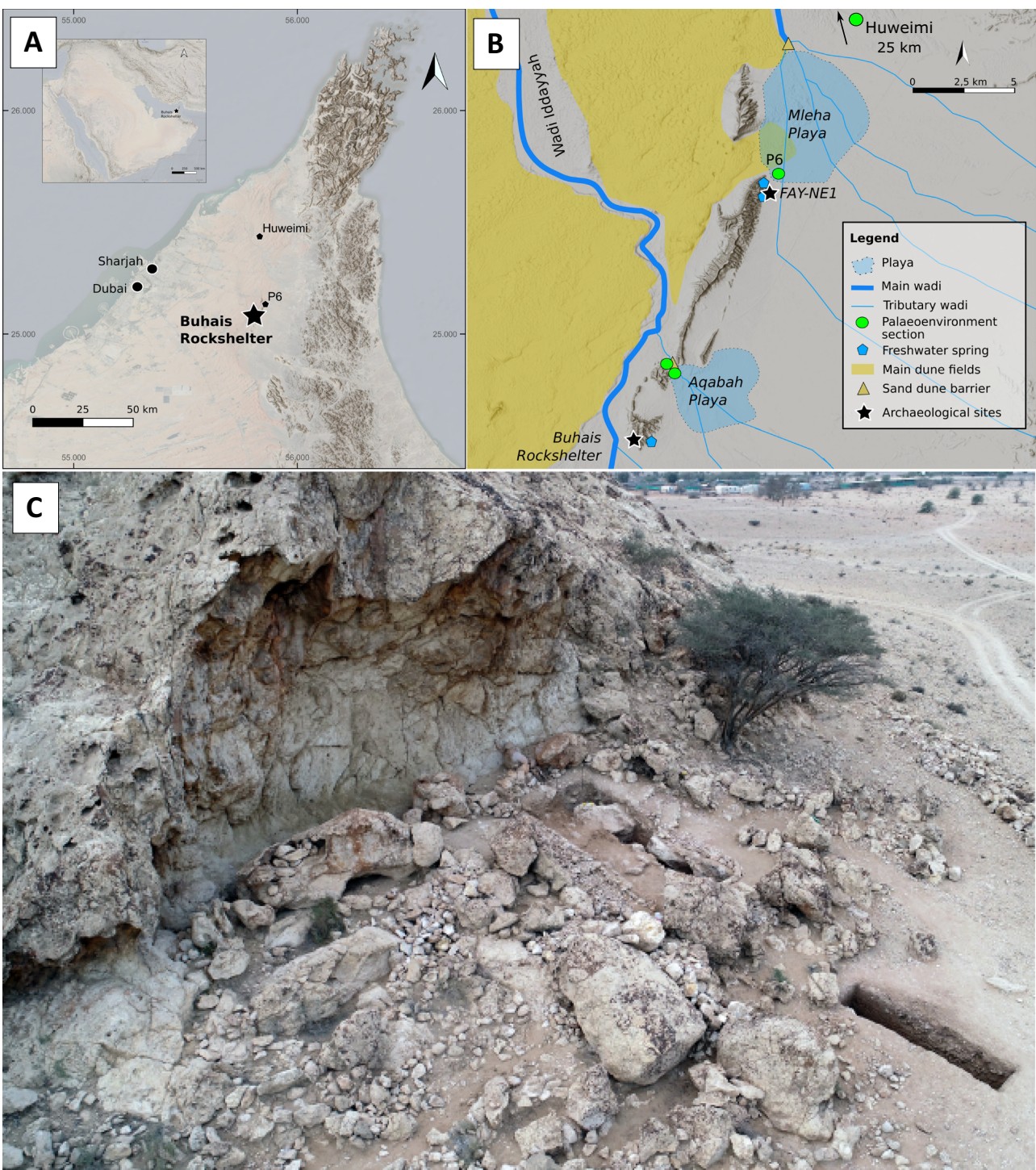

**Fig. 2 | Overview of Buhais Rockshelter. A** Location of the site in Southeast Arabia. **B** location of sites and geographic features within the local environmental setting. **C** Site viewed from the North. Please note that the extension of the playas shown here derive from calculations by Bachofer[18]. Freshwater spring refers to sinter formations that indicate Pleistocene spring activity. Base maps shown in A and B have been developed from public domain data GMTED2010 (courtesy of the U.S. Geological Survey) and ESRI world hillshade (Source: Esri, USGS, NGA, NASA, CGIAR, N Robinson, NCEAS, NLS, OS, NMA, Geodatastyrelsen, Rijkswaterstaat, GSA, Geoland, FEMA, Intermap, and the GIS User Community) using QGIS 3.40.

bedrock. Large rockfall blocks were initially removed from the central part of the site to expose the full excavation area. Within the stratigraphic sequence, four distinct Archaeological Horizons (AHs) were identified. The Pleistocene chronology of the site is based on optically stimulated luminescence (OSL) dating of nine sediment samples collected from within and between the AHs (see Methods below and Supplementary Figs. 10, 11, Supplementary Tables 5, 6).

The lowermost unit, (GH 3), directly overlies the bedrock (Fig. 4, Supplementary Fig. 1), is approximately 30 cm thick, and contains the oldest archaeological horizon AH III. Two OSL samples from this layer yield a mean age of 124 ka ± 6 ka (Buhais 3: 123 ± 10 ka; Buhais 7: 125 ± 9 ka). The overlying sediment, which is devoid of archaeological material, was dated to 95 ka ± 9 ka (Buhais 6). The subsequent unit, (GH 2), is 25–30 cm thick and characterised by abundant gravel sized

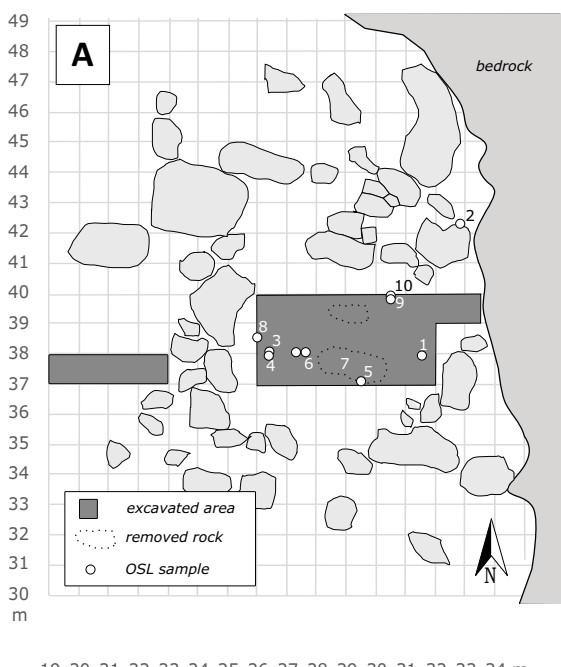

**Fig. 3 | Overview of excavations at Buhais Rockshelter. A** Excavation plan showing the distribution of the excavated trenches. Please note that all archaeological assemblages presented here come from the 20 m² main trench. The remaining trench in the western part was archaeologically sterile. Numbers indicate the location of OSL samples. **B** Cross section along the 40 m north line. Please note the step in the bedrock at about 28 m East. The site's two MP layer are located below this step. Shown grid reflects the local excavation grid.

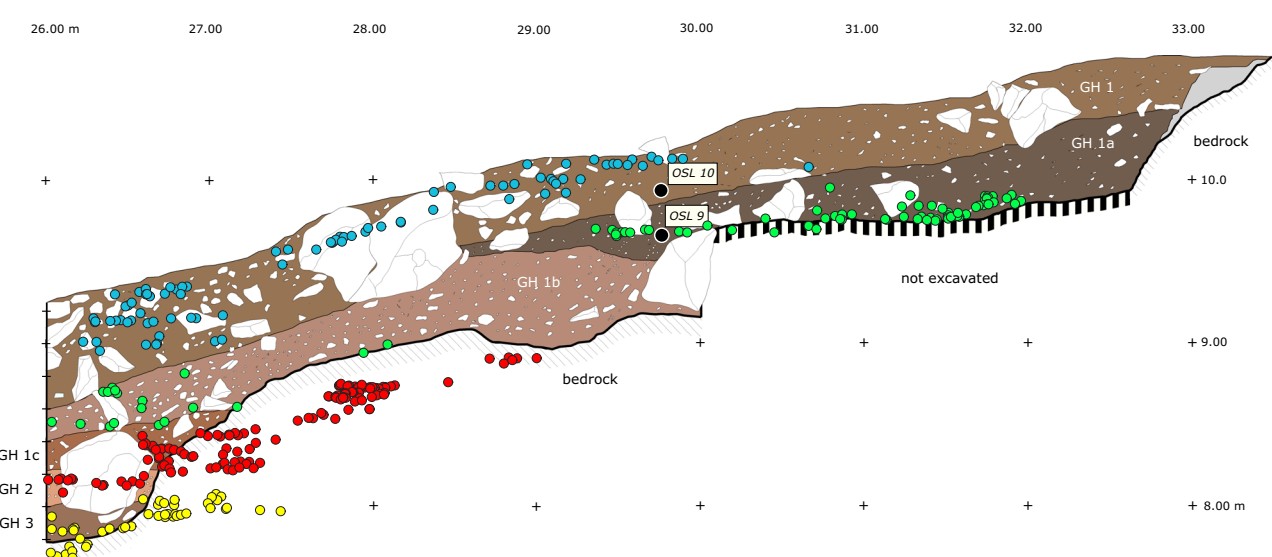

**Fig. 4 | Drawing of the northern profile of the main excavation trench.** Location and extend of geological layers (GHs), archaeological finds (dots) and OSL samples (black circles) are shown. See Supplementary Fig. 1 for the western profile.

clasts (Fig. 4, Supplementary Fig. 1). This sediment is relatively loose and poorly consolidated. GH 2 contains AH II, dated to approximately 60 ka ago (Buhais 4: 59 ± 5 ka).

The overlying layers GH 1d and GH1c form a clearly defined stratigraphic boundary separating GH 2 and GH 1b and thus marking the transition between the site's MP and UP deposits. GH 1b is up to 60 cm thick in the eastern part of the excavation and thins westward to approximately 15 cm. A thin concentration of lithic artifacts was identified within this layer. One OSL sample (Buhais 8) yielded an age of 35 ± 3 ka. GH 1a accumulated behind a large rockfall block, restricting its spatial extent to the eastern part of the shelter near the present backwall (Fig. 4). Two OSL samples from GH 1a indicate a mean

depositional age of 35 ka ± 5 ka (Buhais 1: 30 ± 2 ka, Buhais 9: 41 ± 2 ka). GH 1a contains the lithic assemblage designated AH Ia (see Supplementary Note 1: Site Information for details on the nomenclature). Given statistically indistinguishable ages of GH 1b and GH 1a, artefacts from GH 1b are combined with those from GH 1a and presented as a single archaeological horizon (AH Ia). GH 1 represents the uppermost Pleistocene layer at Buhais Rockshelter and was partly exposed during the Iron Age excavations conducted in the 1990s. The central part of the Palaeolithic excavation was initially sealed by large limestone blocks resulting from overhang collapse. Following their removal, lithic artifacts in undisturbed position belonging to AH I were recovered directly beneath. GH 1 is approximately 40 cm thick, increasing to

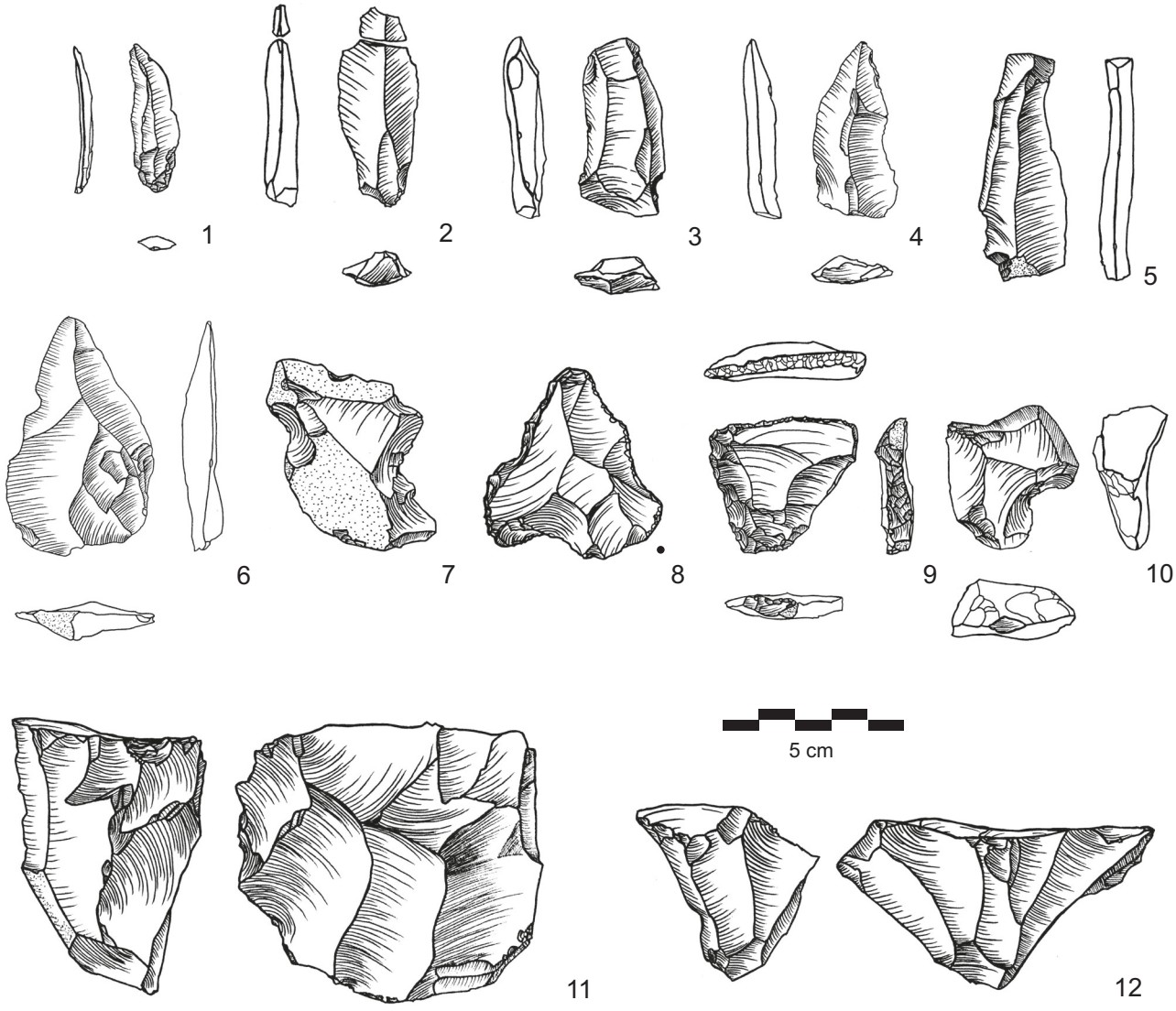

**Fig. 5 | Examples of lithic artifacts from AH II.** 1-5: blades, 6: pointed flake, 7,10: notches, 8: déjété point, 9: short atypical endscraper, 11,12: cores. (Drawings by SK).

about 50 cm in the western part of the excavation. Two OSL dates from this unit indicate a mean depositional age of 16 ka ± 2 ka (Buhais 5: 14 ka ± 1 ka, Buhais 10: 17 ka ± 2 ka).

## Middle and Upper Palaeolithic lithic assemblages

The lowermost AH III comprises a relatively small lithic assemblage ($n = 213$), which nevertheless exhibits typical local Middle Palaeolithic (MP) characteristics (Supplementary Fig. 2). Lithic production in AH III is geared towards flake production (Supplementary Table 1). Levallois cores with unidirectional, bidirectional and centripetal preparation were identified. There is also evidence for the presence of bifacial reduction. The few retouched artifacts include tool types such as notched flakes and sidescrapers as well as one small early-stage biface (Supplementary Fig. 2). AH III resembles typo-technological characteristics known from Jebel Faya assemblages AH VII, AH VI and Assemblage C[13,14].

The LMP assemblage of AH II contains 1039 artifacts. The target of lithic production in AH II was triangular morphologies ranging in terms of elongation from short, wide triangular flakes to large blades with converging edges (Fig. 5). All flake and blade types show relatively large bulbs of percussion and relatively thick striking platforms, which indicates the use of hard stone hammers for their production. Cores feature very limited preparation. Considering technological

characteristics (see also Supplementary Figs. 3–7), there is a range of MP flake and blade production systems evident, including semi-rotating laminar and non-Levallois convergent systems. Evidence for classic Levallois reduction systems are notably scarce in AH II. The often large and unprepared striking platforms, straight ventral surfaces and the lack of evidence for Levallois unidirectional-convergent preparation indicates the preference for technological systems distinguishable from traditional Levallois methods for the production of flakes and blades with pointed morphologies. Tool typologies include typical MP types such as notches and denticulates as well as sidescrapers, and short atypical endscrapers (Fig. 5, Supplementary Table 2). The observed characteristics of the AH II assemblage are currently unknown from the stratified records of the region and represent to our knowledge a new type of MP assemblage.

In contrast to the flake assemblages from AHs III and II, the UP assemblage of AH Ia ($n = 525$) is characterized by a significant focus on the production of elongated flakes, blades and bladelets (Supplementary Table 1). We observe the first occurrence of true UP blade and bladelet technologies in the sequence and in the region. Diverse technological systems for the production of blades and bladelets are evident, including broad face core reduction and bladelet production from cores on flake (Fig. 6). Blade production was carried out using unidirectional-convergent reduction of single platform cores.

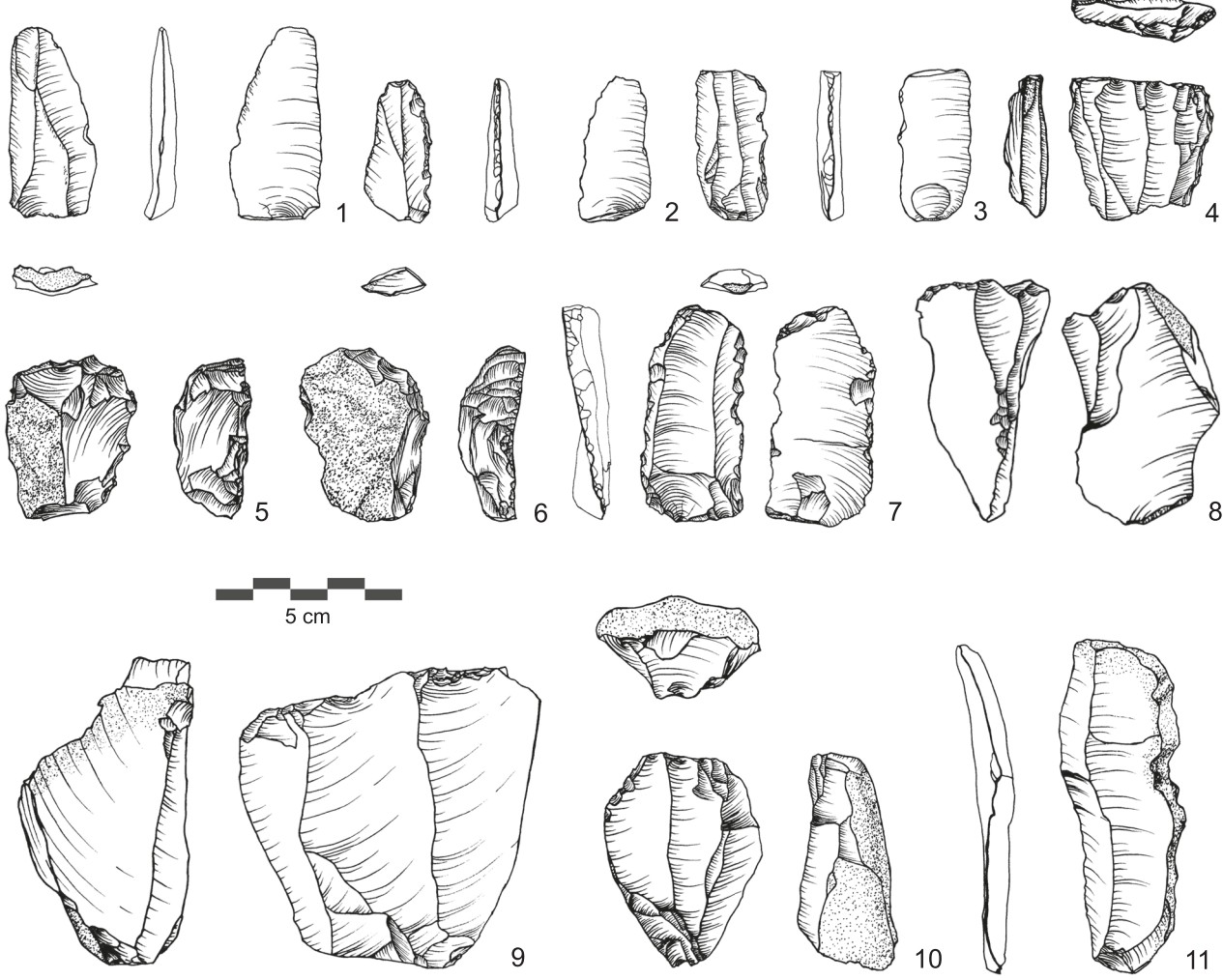

**Fig. 6 | Examples of lithic artifacts from AH Ia.** 1-3: blades, 4: bladelet core, 5-6: carinated scrapers, 7: truncation, 8: multiple burin, 9-10: blade cores, 11: débordant blade. (Drawings by SK).

Unidirectional parallel reduction of cores with two opposed striking platforms was an additionally observed reduction strategy for blades. Striking platforms are often plain but show grinding of edges. Bladelets were produced intentionally using two independent systems. First, there are small bladelet cores where blanks were produced from the broad face of the core (Fig. 6: 4). Second, cores on flakes typologically classifiable as carinated scraper and multiple burins provide evidence for additional independent bladelet production systems (Fig. 6: 5–6). The tool assemblage comprises typical UP types such as endscrapers, burins and truncations (Supplementary Fig. 3, Supplementary Table 2).

The UP assemblage from AH I ($n = 1229$) shows a focus on blade and bladelet production, while flakes as end products play only a minor role (Supplementary Table 1). Production of blades uses the unidirectional-parallel reduction system carried out on single platform cores (Fig. 7). Striking platforms are in general either cortical or plain. Obtained blade blanks are characterized by at least two parallel dorsal ridges, which demonstrates that laminar reduction surfaces were regularly prepared by a series of blade or bladelet removals. Their cross-sections are mainly trapezoidal. Most blade blanks are retouched using different techniques, including truncation of blades with abrupt retouch and backing of blades with semi-abrupt retouch. We recovered several backed bladelets with abrupt/semi-abrupt retouch from AH I (Fig. 7: 5–7). The layer's tool assemblage further includes UP types such as burins, endscrapers, truncations, one unifacial tool and one biface (Supplementary Fig. 3, Supplementary Table 2).

## Palaeoenvironmental research

Archaeological field work in the Faya Palaeolandcape has consistently been accompanied by systematic palaeoenvironmental research[17,18]. To strengthen the basis for interpretations of the palaeoenvironmental context of human occupation in the area during the Late Pleistocene, a 4.70 m deep sedimentary section was excavated at the northern end of Jebel Faya (Fig. 2). Situated within the reconstructed extent of the Mleha playa (following[19]), section P6 offers valuable palaeoenvironmental data. In addition, a 1.70 m deep section was excavated from an interdunal site at Huweimi (section R1) located close to Wadi Dhaid (Fig. 2), into which the Mleha playa catchment feeds as a tributary.

Section P6 consists of a 4.7 m thick stratigraphic sequence characterized by fluvial and alluvial sands and gravels characterising channel flow, as well as sands and silts associated with low energy standing water environments and episodes of aeolian dune deposition (Fig. 8, Supplementary Note 2). Only the results from the Pleistocene part of the sequence are presented here (Fig. 8A, Supplementary Fig. 8, Supplementary Table 3). The lowermost sedimentary unit (470–440 cm; Facies 7, Fl) is composed of carbonate-rich, cemented sands and silts. Small, containing minor ophiolitic gravel clasts, up to 5 mm in diameter concentrated near the top of the unit. High calcium (Ca) and Ca:Si ratios (Fig. 8A, Supplementary Fig. 3) indicate an increased input of carbonate material. This facies likely formed from fine sediments settling out of suspension in a calm lake environment, suggesting low-energy conditions during the existence of the Mleha

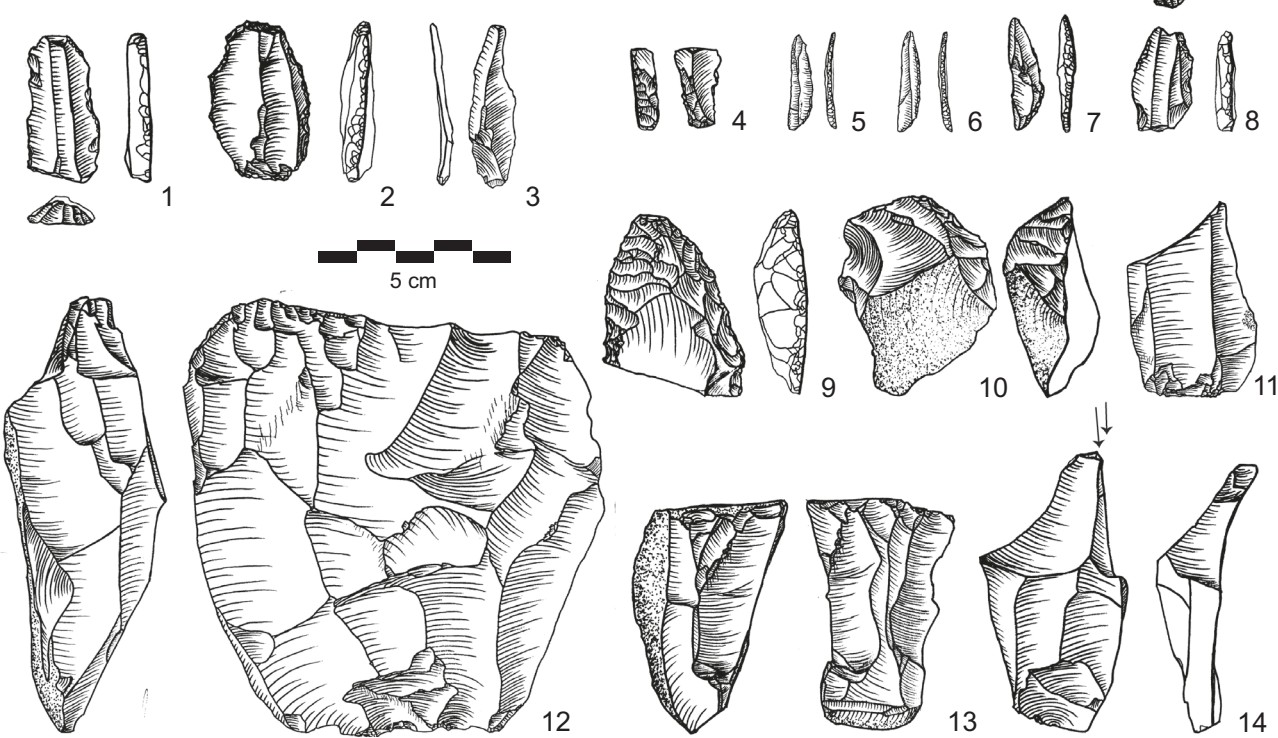

**Fig. 7 | Examples of lithic artifacts from AH I.** 1,8: truncation, 2: endscraper, 3: technical blade, 4-7: backed bladelets, 9: unifacial tool,10: carinated scraper, 11: broken blade, 12,13: blade cores, 14: multiple burin. (Drawings by SK).

playa lake. This unit was dated to >152 ± 14 ka (P6/2023-OSL1). Between 430–365 cm, this unit is overlain by a gravel-rich layer dominated by harzburgite clasts characterised by high concentrations of chromium (Cr), nickel (Ni), iron (Fe), and elevated (Rb+Sr)/Ti ratios (Fig. 8; Supplementary Fig. 8). The gravels are clast-supported, cemented by hydromagnesite, and include pebbles and occasional small cobbles within a sandy matrix. The base of this layer is marked by a sharp, erosional boundary. Slight changes in sand content cause subtle colour differences throughout the sequence. These gravels are interpreted as the result of high-energy, unidirectional fluvial flows transporting sediment from the Hajar Mountains. They likely represent channel lag deposits formed during periods of increased water discharge within wadi channels and adjacent areas. A sample from 370 cm yielded an age of 59 ± 7 ka (P6/2023-OSL2).

From 365–330 cm, the sequence is dominated by a widespread red aeolian layer (Facies 8, Sem), consisting of up to 98% fine to medium sand. This unit represents wind-blown dune material transported from the Rub al-Khali sand sea to the west of Jebel Faya. The gravel layer below it, rich in harzburgite clasts, is geochemically distinct and marked by high values of Cr, Ni, Fe, and (Rb+Sr)/Ti − a clear geochemical fingerprint of its source in the Hajar Mountains. In contrast, this overlying aeolian unit shows a sharp drop in those elements, along with a significant increase in Si and the K/Ca ratio (Fig. 8, Supplementary Fig. 8). The sediments are moderately to well sorted, with sub-rounded to well-rounded grains, and feature faint pin-striped bedding. A sample from this layer was dated to 39 ± 5 ka (P6/2023-OSL3).

Above this, from 330–275 cm, lies another gravel layer (Facies 1, Gcm), composed of clast-supported, hydromagnesite-cemented gravels dominated by harzburgite clasts. This unit is geochemically distinct, with high levels of Cr, Ni, Fe, and (Rb+Sr)/Ti, and lower values of Si and the K:Ca ratio (Fig. 8, Supplementary Fig. 8). The gravels are interpreted as the product of high-energy flows during periods of increased water discharge. The base of the unit is marked by a sharp, uneven erosional contact with the underlying aeolian sands. An OSL

date from this layer indicates an age of 30 ± 2 ka (P6-13). The sediments above this unit are of Holocene age (Fig. 8A) and are not discussed here.

The sediment sequence at Huweimi (Fig. 8C, Supplementary Fig. 9; Supplementary Table 4) records a shift from aeolian to lacustrine environments during the Late Pleistocene. The basal unit (R1–1) consists mainly of coarse-skewed aeolian sand (~80%), indicating wind-driven deposition typical of interdune settings. Above this, finer sediments (R1–2) dominated by silt and clay were deposited in a low-energy lake environment, reflected by symmetrical to coarse-skewed grain size distributions. Increasing iron and strontium concentrations through this lacustrine unit suggest changing sediment sources or geochemical conditions. Subsequent units (R1–3 and R1–4) show a gradual increase in sand content and decrease in iron, indicating a transition toward more energetic conditions and coarser sediment input.

The uppermost units (R1–5 and R1–6) mark a return to aeolian dominance, with sand content rising from 20% to 75%, reflecting renewed wind activity and likely drier conditions. OSL dating places the basal aeolian sands at around 16 ± 1 ka (HWM-2; Supplementary Table 6) and the overlying lacustrine sediments at 17 ± 1 ka (HWM-1), indicating rapid sedimentation and a brief lake phase between two aeolian episodes. This pattern correlates with regional Late Pleistocene fluvial activation in Wadi Iddayyah (~17–16 ka), highlighting dynamic environmental changes driven by climatic fluctuations during this period[18].

## Discussion

The terminal Pleistocene from about 70 ka until the end of the last glacial period at about 12 ka is globally a period of decreasing temperatures, increasing ice volume in the high and mid-latitudes and significantly changing sea levels[20,21]. In sub-tropical regions such as the Arabian Peninsula, this period is often characterized by decreased precipitation and increased aridity[22,23]. The lack of speleothem growth between about 78 ka and 11 ka identified in Hoti Cave[24] as well as

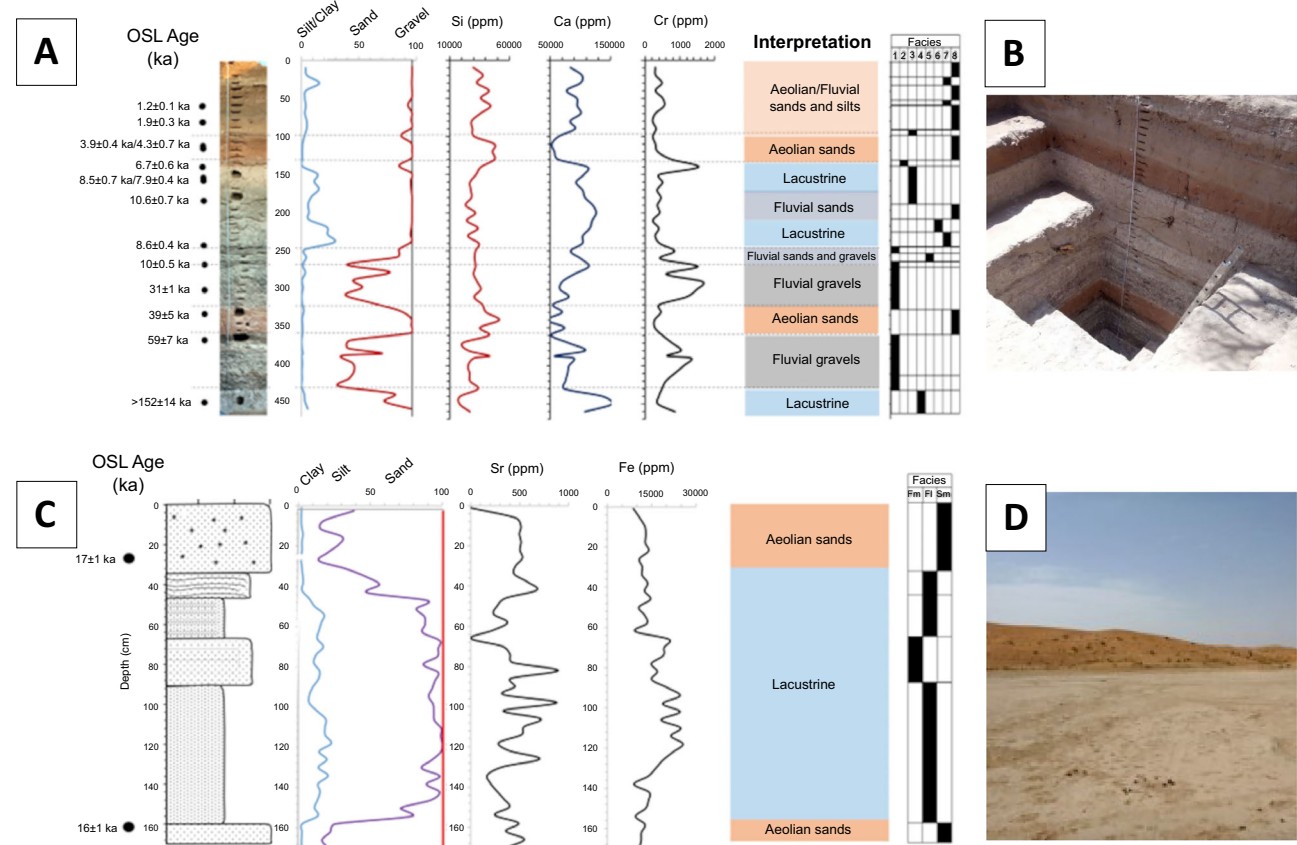

**Fig. 8 | Overview of palaeoenvironmental sections. A** Mleha P6 section with OSL chronology (n = 14), grain size, geochemistry and environmental conditions with facies; **B** view at the main outcrop investigated. **C** Huwemi R1 stratigraphy with facies, OSL chronology and physical and geochemical properties (bottom). **D** View of interdunal playa with dunes in background.

aeolian deposition in the Wahiba Sands between MIS 4 and MIS 2[25,26] provide evidence for arid conditions in SE Arabia at the end of the Pleistocene (Fig. 9). This in addition to scarce archaeological records from this period, often leads to the assumption of a largely uninhabitable Arabian Peninsula between approximately 60 ka and 16 ka[2,27,28]. However, evidence for brief phases of rapid warming identified in Greenland ice cores (Dansgaard−Oeschger events) and a high-latitude to tropical coupling of climate change indicate significant fluctuations of climatic conditions during the last glacial period in many parts of the world[29–31]. Evidence for brief phases of favourable conditions during the terminal Pleistocene is known from the Faya Palaeolandscape[17,18,32,33]. Our palaeoenvironmental results from Mleha site P6 and Huweimi R1 complement these existing records and underline the presence of an arid landscape shaped by multiple brief phases of increased water availability linked to the reactivation of freshwater springs and hydrological systems in the region in particular at 65−57 ka[17,33], 43−34 ka and 20−15 ka[18] (Fig. 9).

The timing of favourable conditions in SE Arabia as indicated by these local records is consistent with results derived from global models of climate change[34,35]. Periods of favourable conditions and modelled human presence in Arabia include 107−95 ka, 90−75 ka, 60−47 ka and 45−30 ka[36]. Archaeological finds demonstrate human presence in Arabia for all of these phases[2–4,7,12,37,38]. This evidence, however, is distributed over large parts of Arabia and the different areas often contain diachronically limited records (Figs. 1, 9). The Faya Palaeolandscape provides deeply stratified archaeological sites with evidence for multiple occupation phases between ca. 210 ka and ca. 16 ka from an area of about 25 km². This includes the three youngest phases mentioned above (90-75 ka, 60-47 ka, 45-30 ka) and evidence

for an occupation phase so far unknown from most parts in Arabia at ca. 16 ka. The two youngest Palaeolithic occupation phases in the Faya Palaeolandscape (Buhais Rockshelter AHs Ia and I) occur just before and just after the Last Glacial maximum (ca. 20 ka), a phase often considered as hyper arid (MIS2, MIP2[14]). Our palaeoenvironmental records from the Faya Palaeolandscape indicate that occupation represented by AHs Ia and I coincide with activation of the Iddayyah fluvial system[18] and development of lacustrine conditions at Huwemi R1 (Figs. 8, 9). This is in line with model results that indicate a phase of favourable conditions and potential human presence in Arabia between 25 ka and 15 ka[39].

The geographic origin of populations inhabiting Buhais Rockshelter at the end of the Pleistocene is difficult to estimate, given the lack of aDNA information from Arabian prehistoric sources. Traditions in the production of lithic artifacts provide a source of information for potential spatial connectivity of SE Arabian populations. Buhais Rockshelter extends the Middle Pleistocene to early Late Pleistocene record (ca. 210−80 ka) from Jebel Faya (site FAY-NE1[13,14,37]), located about 15 km to the north (Fig. 2B). The Palaeolithic sequence at Faya terminates with AH II, which has been dated to ca. 80 ka[37]. This youngest MP assemblage is characterized by a preference for the production of elongated flakes and blades with parallel edges using Levallois bi-directional systems. It represents the end of a locally continuous technological development within the range of the Levallois system during MIS 5[40]. Buhais Rockshelter now indicates that the re-occupation of the area at ca. 60 ka was linked to a technological discontinuity. The lithic assemblage from Buhais Rockshelter AH II shows that Levallois is no longer the predominant technological system to produce stone tools. Instead, systems including semi-rotating

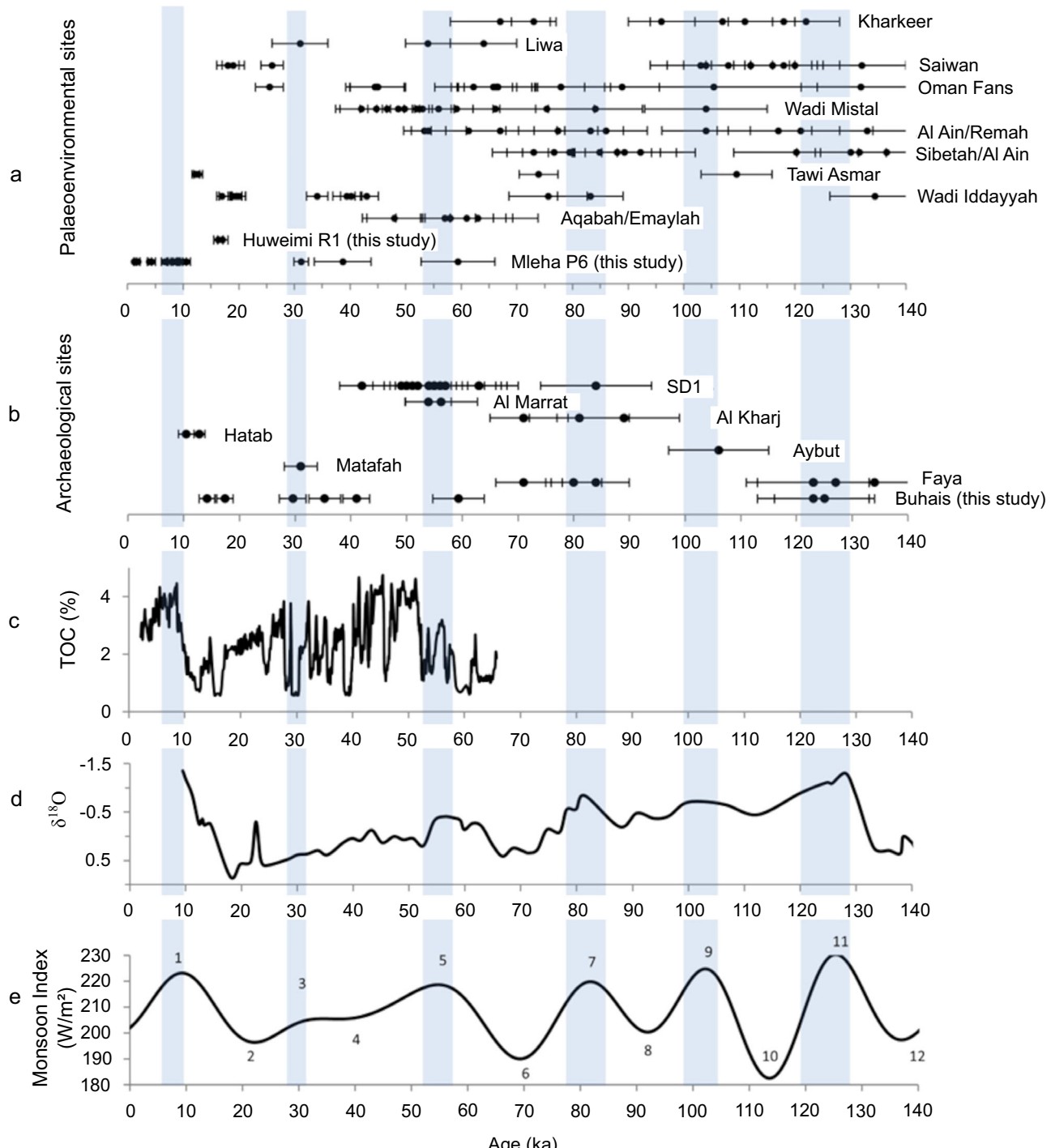

**Fig. 9 | Chronometric context of archaeological and palaeoenvironmental records discussed in the text. a** palaeoenvironmental records 3, 16,17,21-26, 28. **b** dated archaeological sites from Arabia 3,6,12,13,27,46. **c** Arabian Sea core 136 KL TOC (%) 32. **d** Arabian Sea Neogloboquadrina duterti δ18O33. **e** Indian Ocean Monsoon Index 34. Numbers denote Monsoon Index Peak reference system 14. Grey bars indicate precessional insolation maxima. Dots shown in a and b represent single chronometric age estimates with their errors at one sigma (whiskers).

laminar and non-Levallois convergent systems replace the previously dominant Levallois[40].

The observed break in the technological sequence of the Faya Palaeolandscape together with a significant gap in the archaeological record between 80 ka and 60 ka, is consistent with models suggesting hostile environmental conditions in Arabia at this time[36]. The Toba super-eruption at 74 ka could be a key factor when considering potential causes for the observed discontinuity between 80 ka and 60 ka. This volcanic eruption is thought to have led to significant deterioration of climatic conditions and a bottleneck in the global human population[41]. In SE Arabia this could have led to the local extinction of modern human populations that thrived in the region since their early colonization between 210 and 170 ka[14]. The distinct lithic technologies observed in the layers pre- (Faya AH II) and post-dating (Buhais AH II) the age of the Toba event would support the conclusion of discontinuous occupation in SE Arabia between 80 ka and 60 ka. Such a discontinuity would contrast with claims for a continued settlement in SE Arabia between 80 ka and 50 ka and the

existence of an Arabian Standstill population[42]. Our empirical data currently does not support this model. Instead, our results are in line with models delineating an occupation gap and potential population bottleneck in that time frame[36,41].

Assemblages from the Arabian Peninsula contemporaneous with Buhais Rockshelter AH II (ca. 60 ka) are scarce and include Shi'bat Dihya-1 (SD1) in Yemen[3] and Al Marrat 3 (ALM 3) in the Nefud Desert in northern Saudi Arabia[7] (Figs. 1, 10). The Buhais AH II assemblage demonstrates technological similarities with SD1, including a prevalence of non-Levallois methods, unidirectional production of morphologies ranging from short, wide flakes to blades with converging edges and minimal core preparation[3]. The lithic assemblage from ALM 3, in contrast, features a dominance of the Levallois concept, a focus on triangular points and reduction strategies involving an intensive preparation of cores such as the faceting of striking platforms[7]. Technological characteristics observed at ALM3 correspond with evidence from Levantine LMP assemblages[7] and indicate a connection to the north. Considering these technological patterns, we hypothesize that human populations inhabiting SE and SW Arabia about 60–50 ka shared a common cultural background, which was independent from northern Arabia (Fig. 10).

A second LMP technocomplex in SW Asia, the so-called Zagros Mousterian technocomplex[43], is also characterised by an emphasis on Levallois systems and intensive preparations during the lithic reduction process. The Zagros Mousterian has been identified mainly in northern parts of the Zagros Mountain range and occurred between about 130 ka and 40 ka[44]. None of the typo-technological characteristics of the Zagros Mousterian have been observed in the MP assemblages of the Faya-Buhais region suggesting a chronologically deep divide between cultural developments in the northern Zagros and SE Arabia. In the southern Zagros Mountains at sites such as Ghar-e Boof (Fig. 10), however, there is evidence for MP lithic traditions that are different from typical Zagros Mousterian. Ghar-e Boof Cave documents MP occupation around 80–70 ka in AH VI and 60–50 ka in AHs Va-c[45]. Studies of the lithic assemblages indicate only a minor role of Levallois systems in AHs Va-c[46], which is consistent with observations from southern Arabia. This could indicate that SE Arabia and the southern part of the Zagros Mountain range formed an area of cultural interaction about 60–50 ka ago (Fig. 10). Palaeoshoreline reconstructions indicate that the Gulf basin was exposed at that time[47,48], which would facilitate exchange among populations settling along the coasts and related hinterland.

Whether the non-Levallois dominated assemblages from southern Arabia and the southern Zagros Mountains reflect migration of *Homo sapiens* populations, remains difficult to test. It has been suggested that the Zagros Mousterian was linked to Neanderthals, while in the Levant there is evidence for both Neanderthals and *Homo sapiens* inhabiting the region during the Levantine LMP. Recently it has been suggested that SE Arabia and the central/southern Zagros Mountains were ecologically rather suited to *Homo sapiens* than Neanderthal populations[49]. This would support a conclusion about distinct cultural spaces in northern and southern Arabia about 60–50 ka ago (Fig. 10).

Available palaeoenvironmental data corresponds to an observation of connected areas in the SE and SW of Arabia by providing evidence for an activation of drainage systems in southern Arabia at that time as indicated by the deposition of gravel dated to ca. 54 ka at Al-Quwaiayh in the central Arabian Peninsula[50], alongside the development of interdunal wetlands in the al-Kharkheer[51] and in the Liwa region[52] of the Rub al'-Khali (Figs. 9, 10). The emergence of inland wetlands, which would enable West-East or East-West connections, coincides with evidence for amplified upwelling[53,54] derived from monsoonal activity in Arabian Sea record[55]. Prudence suggests refraining from drawing final conclusions due to still limited number of records, but taking available palaeoenvironmental (P6 and[17,32]) and archaeological data (Buhais AH II, SD1) into account, it seems likely that

at 60–50 ka the exchange of ideas and people among human populations in SW and SE Arabia and along a southern corridor was possible. This conclusion is consistent with results from climate forced human migration models, which suggest a migration wave at that time due to enhanced net primary productivity in Arabia[36].

A second clear technological break in the Faya Palaeolandscape is evident through the 35-ka record (AH Ia) of the Buhais Rockshelter sequence. Flake oriented production systems of the LMP are replaced by UP blade and bladelet technologies. The early UP occupation phase was preceded by a phase of hyper-arid conditions in the region between ca. 41–37 ka (P6 and[13,18]) during the middle part of MIS3 (MIP4[13]), which suggests that demographic factors have caused the observed technological discontinuity in the lithic culture rather than local evolutionary or adaptive processes. Moreover, at Buhais Rockshelter and in Arabia in general, there is currently no evidence for a distinct MP to UP transitional technocomplex akin to the Initial Upper Palaeolithic (IUP), a phenomenon known from many other parts of SW Asia[56,57]. The current archaeological record instead suggests that in SE Arabia at the latest around 35 ka, local LMP traditions are replaced by full-fledged UP lithic traditions without transitional phase. This is a preliminary conclusion based on the current record whose validity must be re-evaluated in the future based on an extended record.

Layers AH Ia and AH I from the Buhais Rockshelter sequence reveal two technologically linked UP occupations. To date, stratified and well-dated records for Pleistocene UP traditions are extremely scarce on the Arabian Peninsula. The only known site with stratified and dated material is Matafah in southern Oman (Fig. 1). Here, excavation of a 3 m² test pit revealed the presence of 89 lithic artifacts larger than 2 cm in an archaeological layer called AH III, which was dated to ca. 33 ka[12]. The assemblage contains no cores and the excavator concludes that the small number of finds precludes further conclusions about technological details[12]. The estimated age in addition to the observation that about half of the recovered lithic artifacts feature blade characteristics, including the presence of eight backed bladelets in Matafah AH III, however, provide arguments to assign this assemblage to an Upper Palaeolithic context. While accepting limited insight from Matafah, observed common technological traits might suggest a connection between southern Oman and SE Arabia for the 35 ka time window (Fig. 10).

Archaeological assemblages and palaeoenvironmental sequences presented here, allow starting to illuminate the UP phenomenon in Arabia. The earliest UP layer (AH Ia) at Buhais Rockshelter demonstrates that UP technology was present in SE Arabia at ca. 35 ka, a time when this phenomenon was well established in most parts of SW Asia[9–11]. One striking characteristic of the Buhais Rockshelter UP is the presence of multiple independent blade and bladelet production systems, a characteristic that defines the Aurignacian phenomenon in Europe and SW Asia. It has been observed in many UP assemblages from the Levant and the Zagros Mountains (Fig. 10) including Sefunim[58], Manot Cave[9], Ksar Akil[11], Shanidar[59], Warwasi[60], Yafteh[61] and Ghar-e Boof[62]. In contrast, independent blade and bladelet production systems, including carinated artifacts, are characteristics unknown from other parts in Arabia[63] and from Africa[64] at approximately 35–30 ka. Consequently and based on currently available records, a connection of the UP in SE Arabia to the Aurignacian phenomenon of SW Asia[59,60,65,66] is possible. This stands in contrast to the directionality observed in earlier assemblages of the Faya Palaeolandscape, which was predominantly oriented to the West, to SW Arabia and Africa. It also contrasts with claims for an expansion wave out of Africa through crossing the Bab-el-Mandeb at 45–30 ka[36] and instead hints at an extension of lithic traditions into SE Arabia from the North/Northeast (Fig. 10). Since the Gulf basin is expected to be exposed in large parts at this time[47,48], a connection would be possible.

Integrating results from Buhais Rockshelter and the Faya Palaeolithic sequence indicates two important changes in SE Arabia during

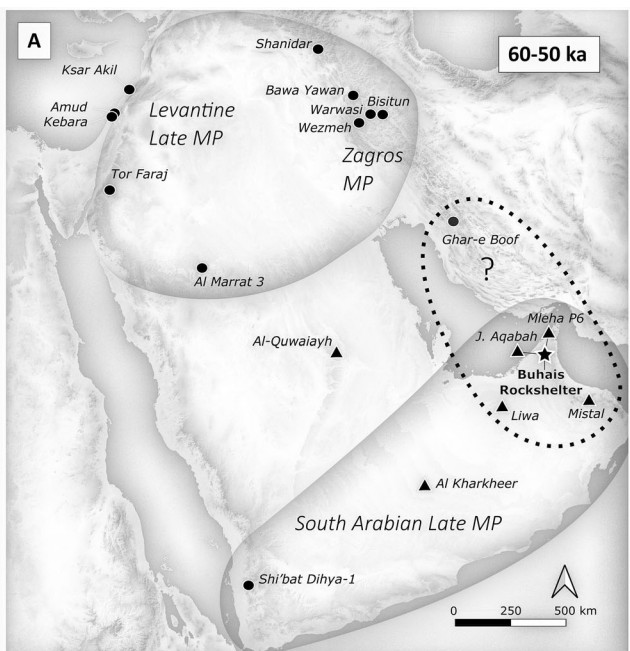
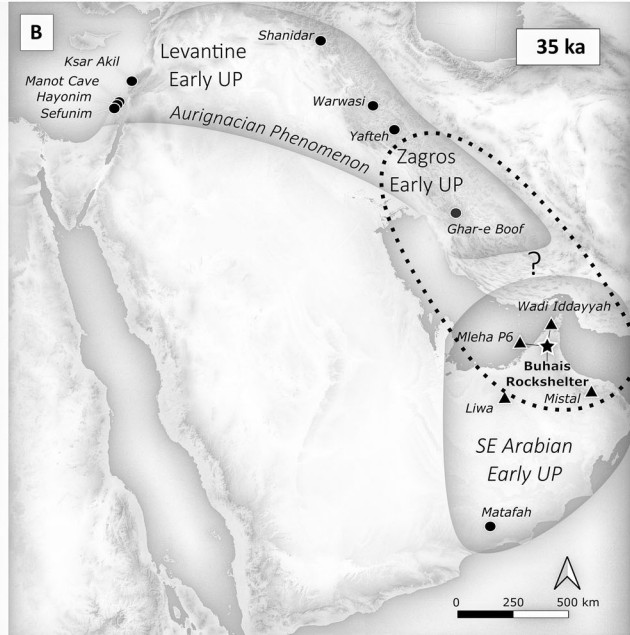

**Fig. 10 | Potential areas of cultural exchange.** Sites and potential connected cultural spaces mentioned in the text for the periods **A)** ca. 60–50 ka and **B)** ca. 35 ka. Please note that during both periods much of the Gulf floor was exposed[47,48]. Base map shown in A and B developed from public domain data GMTED2010 (courtesy of the U.S. Geological Survey) using QGIS 3.40.

the MIS 5 to MIS 2 period. First, a long phase of technological continuity between ca. 170 ka and 80 ka as observed in the Faya sequence (Faya AHs VII-II) is replaced by a period of repeated technological change from about 60 ka onwards (Buhais AHs II, Ia). Second, our archaeological records hint at a change in the spatial connectivity of populations occupying SE Arabia. Between ca. 170 ka and 60 ka, records indicate that the cultural space of SE Arabian populations includes regions in SW Arabia, Eastern Africa and potentially the southern Zagros Mountains, while from 35 ka onwards connections seem to be exclusively towards regions to the North/Northeast (Fig. 10). Does this observation reflect an eastward expansion at 60 ka and the establishment of a stable human population in SW Asia at some time after 60 ka? Did expansion pulses reach SE Arabia repeatedly from there? Such a scenario would be consistent with the proposed Basal Eurasian population, whose geographic core region researchers broadly locate in the southern Gulf region[67]. This genetic phantom population is characterised by scarce evidence for Neanderthal admixture and is thus hypothesised to have split off from an early out-of-Africa population before the admixture with Neanderthals and the differentiation of all other Eurasian lineages[68]. It has been shown that Basal Eurasian ancestry was widespread in ancient SW Asia, which could be the result of a long-lasting presence in the region[68]. Neanderthal admixture has recently been estimated to 45–49 ka, which may imply that all pre-50 ka populations outside Africa did not contribute to the formation of the modern global population[69]. Our results demonstrate that one of the last pre-50 ka populations in SW Asia can be found in SE Arabia at Buhais Rockshelter. Presented chronological and cultural records form an empirical basis that provides a first anchor point for further cross-disciplinary research into deciphering the Basal Eurasian population and human's last glacial history in SW Asia.

## Methods

Archaeological excavations and sampling for palaeoenvironmental and chronometric analyses have been conducted with permission by the Sharjah Archaeology Authority issued to KB. Export permissions for geological samples have been issued to KB and AGP. Handling and storage of finds and sediment samples followed rules and guidelines by the involved institutions.

### Archaeological excavation

Buhais Rockshelter is located at the southern end of Jebel Buhais facing to the west towards the sand dune area, which provides protection from the sun in the morning (Fig. 2). The site provides a sheltered area of about 20 m width and 3 m depth and a potential total habitation area of about 140 m². The big blocks covering large parts of the modern rock shelter area represent repeated roof collapse. These blocks preserved about 1.7-meter-deep sediments. The site is located just east to a small wadi that drains surface water from its catchment area of about 25 ha in southwestern direction towards Wadi Iddayyah (Fig. 2B).

Following modern standard excavation techniques, we set up a local x-y-z grid using a total station to allow piece plotting of all finds and features at <1 cm precision in all dimensions. The grid was roughly north-south oriented with the x-axis being perpendicular to the backwall of the rock shelter and y increasing towards north. The excavated area was subdivided into 1 × 1 m areas, each named after their coordinates in the southwest corner (e.g. square 26/36 had x = 26 m and y = 36 m coordinates in the southwest corner). Excavators dug in ¼ squares of these 1 × 1 m areas following the natural geological horizons. Archaeological (AH) and geological horizons (GH) were defined during excavation. Hence, all finds and all sediment from every bucket, including later finds from the sieving, were assigned to one AH and one GH. Every quarter square was excavated to a depth of about 3 cm, which fills about one 12 l bucket. The exact depth reached with every round of excavation depends on the sediments characteristics as well as slope and extend of the defined GHs. All buckets were sieved using 6 mm and 2 mm meshes to recover the small size fraction of the lithic assemblage. Every find was piece plotted using a Leica total station. Buckets were measured at the centre point of the ¼ squares they were excavated from. The latter allows the small finds that were recovered during the sieving of the buckets to be located within the quarter meter and at the correct height. All measurements are recorded in a database during the

excavation using the EDM software kindly provided by S. McPherron (https://github.com/surf3s/EDM).

## Palaeoenvironmental research

Sedimentological samples were collected from a 4.7 m vertical sample trench profile P6 excavated in the Mleha Playa (Fig. 8, Supplementary Table S3). The sequence comprised fluvial gravels, fluvial sands and silts and aeolian dune/sands. Lithofacies were described and differentiated based on sedimentological characteristics and sedimentary structures following different classification systems[70,71]. Sediment samples were systematically collected from the open trench face and placed in zip lock bags.

Huweimi (Fig. 8) is located in the dune field 3.5 km north of the village of Ar Rashidiyya. This area comprises a series of small interdunal playa basins separated by dune ridges, which are up to 20 m above the playa floor areas. The sampling site was found within an interdunal corridor, which extended for 1.5 km x 0.5 km. On the eastern flanks of the sub-basins, lake sediments have been eroded by gullies to form a series of low, muddy yardangs. These are up to 1 m high and 5 m in length. The western sections of the sub-basins have been eroded and deflated. A 170 cm deep sample pit was excavated and samples collected at 4 cm intervals.

Sediment bulk density was measured using calibrated brass pots following standard procedures[72]. Organic, carbonate and inorganic contents were determined using the loss on ignition (LOI) method at 550 °C and 950 °C. Magnetic susceptibility was measured with a Bartington MS2 meter and MS2 sensor, with samples oven dried at 40 °C, weighed to two decimal places and measured in replicate. Values are reported as the mass specific susceptibility ($\chi_{lf}$) measured as $10^{-6} m^3 kg^{-1}$. For particle size analysis, samples were oven-dried at 105 °C for 12 h, then disaggregated and sieved at 2 mm, 1 mm, 0.5 mm, 0.25 mm, 0.125 mm, and 0.063 mm. Additional grain size distributions (0.02–2000 µm) were obtained by laser diffraction using a Malvern Mastersizer 2000 after overnight soaking in a 5% sodium hexametaphosphate solution. Geochemical analyses of major and trace elements were performed non-destructively using two portable X-ray fluorescence (pXRF) devices: an Olympus Vanta (GeoChem calibration, 8–40 kV, automatic filter selection) and a Niton XLT Series pXRF. Elements including Cr, Ni, and Fe were used to trace fluvial input from ophiolite-rich sources in the Hajar Mountains, while Ca indicated allochthonous carbonates and Si represented aeolian silts and sands. Geochemical ratios were used to interpret sediment provenance and weathering intensity: Ca/Si to distinguish between autochthonous and allochthonous carbonate[73,74]. Rb/Sr ratio to assess weathering and terrigenous inputs[75]; (Rb + Sr)/Ti as a grain-size independent proxy for chemical weathering[76]; and K/Ca to evaluate terrigenous vs lacustrine trends, with elevated K-values linked to clay minerals or K-feldspars transported by fluvial or aeolian processes/aridification[77]. The full sedimentological and geochemical dataset is detailed in Supplementary Figs. 8 and 9.

## Age determination

We used the optically stimulated luminescence (OSL) technique to date the Buhais Rockshelter sequence and the palaeoenvironment sections. OSL determines the last exposure to sunlight of quartz minerals from the sediments that contain the archaeological records (see Supplementary Note 3: Age Determination for details). Considering the archaeological sequence at Buhais Rockshelter, we systematically sampled the sequence during the course of the excavation in different areas within the excavation in 2018, 2019 and 2022 (Fig. 3A). Key strata such as the UP-containing GHs 1 and 1a have been repeatedly sampled from different parts of the excavation to ensure validity of the results. In total we collected ten samples from Buhais Rockshelter. Sample Buhais 2 has been excluded here since it is from a stratigraphic position above the sequence presented here linked to the Holocene occupation of the site. All produced OSL results are in chronological order and correspond with the sequence of layer deposition.

Considering the palaeoenvironment section P6, we collected in total 14 samples covering the entire 4.7 m deep section (Fig. 8). All samples were shipped to and processed in the OSL dating lab of the University of Freiburg.

Sampling for OSL dating was carried out in the excavation by collecting material through excavating at night using red headlights. First, the outer, light-exposed layer of about 5–10 cm was removed. The sediment extracted from the expose was sieved on-site (< 2 mm) to substantially reduce the amount of material for shipping, which was transferred into opaque bags. For the palaeoenvironment sections, opaque tubes were hammered into the sediment and transferred into black bags. A representative sample of about 200 g from the surroundings of each OSL sample was collected for high-resolution gamma spectrometry (no sieving). The advantage of the latter approach is that it allows for the detection of radioactive disequilibria in the Uranium decay chain[78–80]. Such disequilibria can occur in sediments rich in carbonate, as the result of post-deposition mobilization related to percolating water. A loss or gain of isotopes from the Uranium decay can produce significant age offsets.

The radiation dose accumulated by the quartz grains since deposition ($D_e$) was determined for small (1 mm) aliquots of quartz (ca. 50 grains), very close to a single-grain level. About 40–50 aliquots were measured per sample for archaeological layers, and ca. 20 for most palaeoenvironmental sections. $D_e$ distributions were analysed applying different statistical methods as explained in the Supplementary Note 3.

## Reporting summary

Further information on research design is available in the Nature Portfolio Reporting Summary linked to this article.

## Data availability

All data used in this paper are provided in the main text and supplementary information. Archaeological samples used here are stored by the Sharjah Archaeology Authority (SAA). Requests for access should be addressed to SAA (info@saa.shj.ae).

## Code availability

Luminescence data analyses were performed using functions implemented in the R package 'Luminescence' 1.1.2.

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

## Acknowledgements

We are grateful to HH Sheikh Dr. Sultan bin Muhammad Al Qasimi and the Sharjah Archaeology Authority for permission and continued support of this project. This project received funding from the Deutsche Forschungsgemeinschaft through a grant to KB (BR 5562/6-1). S.K. would like to thank the Collège des écoles doctorales de l'Université Paris 1 Panthéon-Sorbonne and the France Excellence program implemented by the Embassy of France in Korea for receiving funding. We would like to extend our sincere gratitude to the Buhais Rockshelter excavation teams, especially Alexander and Anika Janas. We would also like to thank Rémy Crassard, Anne Delagnes, Sylvain Soriano, Solène Denis, Roxane Rocca and Boris Valentin for sharing their expertise on lithic technology.

## Author contributions

K.B., F.Pr., S.J., E.Y. and A.G.P. designed this study and organized and conducted fieldwork and sampling. K.B., S.K., F.Pr., G.W.P., F.Pa. and A.G.P. carried out sample preparations and subsequent analyses. K.B., S.K., F.Pr. and A.G.P. prepared the manuscript, which all co-authors have reviewed.

## Funding

## Competing interests

The authors declare no competing interests.
