## [Transparent Peer Review file · Nature Communications]

Evidence from Buhais Rockshelter for human settlement in Arabia between 60,000 and 16,000 years ago

Corresponding Author: Dr Knut Bretzke

Version 0:

Reviewer comments:

Reviewer #1

(Remarks to the Author)

Comments on the manuscript "Evidence from Buhais Rockshelter reveals human settlement in Arabia between 60,000 and 16,000 years ago" by Bretzke et al.

This study reports OSL-based human occupations at ~125, 59, 35, and 16 ka at Buhais Rockshelter (Arabian Peninsula), challenging the previous Last-Glacial abandonment models and reframing Arabian habitability and human dispersal. It fits the scope of the NC, and could have wide readership. I will suggest major to moderate revision before publication.

My review centers on two pillars: (1) the robustness of the chronological framework that underpins all archaeological and palaeoenvironmental interpretations, and (2) the clarity and readability about field-wide significance and cross-disciplinary relevance expected by NC.

First, the authors should strengthen validation of the OSL ages:

1. The study's chronology places substantial weight on statistical age models (MAM/FMM). While the authors explain key parameter settings (e.g., σ_B), these models rest on specific assumptions and empirically set inputs. Model application should be guided not only by statistical grounds but also by the depositional context of each sample and any additional pertinent information (e.g., independent age control). Regarding De analysis, two major concerns need to be addressed:

Enhance geomorphological justification for age model application. For example, the authors substitute MAM with FMM for sample BUHAIS3, hypothesizing three distinct grain input processes: aeolian influx (well-bleached), rock face weathering (unzeroed), and post-depositional mixing (younger mixing). While this reasoning is plausible, the processes proposed, e.g., weathering and bioturbation leading to discrete components, require dedicated discussion and empirical support from sedimentary or geomorphological evidence.

The absence of independent age control for cross-validation constitutes a significant weakness in the chronological assessment. The author should acknowledge this limitation or integrate independent chronological controls (if available) for validation.

2. For the Dr calculation, the authors should provide a clear and evidence-based justification for the assumed water content values (e.g., comparison with regional moisture measurements, soil properties, or saturation limits used in similar regional studies) or explain the rationale for the assumed ranges ($3\pm 3\%$ and $4\pm 4\%$) – how did you integrate climate change and geological history into this value/error? What was the field water content in comparison?

Second, while the manuscript's technical rigor is strong and detailed, the Discussion's through-line seems fragment, diluting the translation of the chronological results into the narrative readability for capturing new noteworthy findings and perspectives. The through-line thereby requires strengthening.

Clarify a stepwise pathway for challenging Last-Glacial absence models: why the consensus held the interior to be unpopulated in hyper-arid periods (MIS 4/3/2), how stratigraphic scarcity and regional climate interpretations shaped that view, and how your 60–16 ka chronology reframes it via multi-pulses of terminal-Pleistocene occupation with short-lived hydrological conditions.

Once the core findings are sharpened, the compilation of dated archaeological sites should enable revisions to dispersal routes and late-Pleistocene settlement dynamics, which could be delineated further in both text and map. For regional comparison with "northern corridor" frameworks, a similar work of OSL-based MIS 5 Levant wetland corridors by Abbas et al.

(2023, Human dispersals out of Africa via the Levant, Science Advances) would be helpful. Make the cross-disciplinary bridge explicit by showing how the Buhais sequence serves as an archaeological anchor for evaluating the hypothesized Basal Eurasian population; temper claims where evidence is indirect.

A few missing references with OSL ages of Arabian water-lain sediments in the last glacial:

1. Steffen Mischke et al., 2021. A Late Pleistocene Wetland Setting in the Arid Jurf ed Darawish Region in Central Jordan. *Frontiers in Earth Sciences*. doi: 10.3389/feart.2021.722435
2. Bety S. Al-Saqarat, et al., 2020. A wetland oasis at Wadi Gharandal spanning 125–70 ka on the human migration trail in southern Jordan. *Quaternary Research*, <https://doi.org/10.1017/qua.2020.82>
3. Abbas M et al., 2023. Human dispersals out of Africa via the Levant. *Science Advances*, <https://www.science.org/doi/10.1126/sciadv.adi6838>

Reviewer #2

(Remarks to the Author)

This manuscript reports new information from the Buhais Rockshelter in the United Arab Emirates. While the work reported is scientifically robust and has the potential to be significant, it comes across largely as a site report and it does not contribute a substantive, novel, argument or testing of hypotheses. For instance, take the title, 'Evidence from Buhais Rockshelter reveals human settlement in Arabia between 60,000 and 16,000 years ago'. Given that several sites are already known in this time period in Arabia (e.g. SD-1, Matafah), what is fundamentally novel about the new findings? The most important part of the study is perhaps the discovery of Upper Paleolithic material – yet the authors say they are working on a separate paper about this (401), so not much information is given.

Care is also needed with strawman arguments, namely they claim to “contradict the prevailing view of human absence in Arabia at the end of the Pleistocene”. The reference given for this is a paper about a speleothem record from Yemen. That paper argues wet periods facilitated dispersals into Arabia, but it is a strawman argument to claim that this study in particular, and the ‘general view’ more broadly, is that there were no humans in Arabia at the end of the Pleistocene. The work of Rose, Hilbert etc has long argued for a terminal Pleistocene human presence in southeast Arabia.

So while the study reports important findings, it does not seem like these are presented in a particular useful way, it is more like a site report with some vague indications of how the findings relate to things like adaptation to arid conditions and ‘distinctive patterns of settlement dynamics’ (page 3). Fundamentally, the story is seemingly one of repeated migrations into the area in association with wet periods. The authors argue against Late Pleistocene population continuity in Arabia. Instead, it looks like they have identified pulses of hominin dispersal into Arabia, largely correlating with those already known (e.g. SD-1 ca 60 ka, Matafah ca 33 ka). They are also present a somewhat contradictory argument by attacking the point that dispersals into Arabia correlate with wet phases, by saying their studies show the evidence of people in southern Arabia “during the presumably arid last glacial period”, but directly contradict themselves by finding that actually the periods of occupation correlate with “repeated reactivation of freshwater springs and hydrological systems”. What their findings suggest is that the European-derived glacial-interglacial dichotomy may not be the most useful framework for low latitude palaeohydrological change – as indeed many papers have argued in recent years. They themselves point out how actually the regional sequence of SE Arabia suggests human presence in association with Monsoon Index Peaks, when rainfall increased in the area.

The ‘nuts and bolts’ of the paper are good, such as the figures and the general descriptions of the deposits, their associated archaeological assemblages, and luminescence age estimates are all clearly presented and logical. The problem is the lack of orientation of the study to the key research themes in the region.

Reviewer #3

(Remarks to the Author)

The role of the Arabian Peninsula in the expansion of anatomically modern humans is a hotly debated topic in archaeological and palaeo-anthropological research. The “Southern dispersal route” is suggested by genetic evidence, but sound archaeological evidence is missing up to now. Therefore, any data from this region and this period of time is most welcome. Especially the absence of stratified deposits, covering the transition from Middle Palaeolithic to Upper Palaeolithic stone artefact traditions during the Late Pleistocene made any conclusions about this important step in human history impossible. Therefore, Buhais Rockshelter is a key-site with supra-regional relevance. The presented results are of utmost interest for the scientific community and an appropriate publication absolutely desirable. This leads me to my main point of criticism. In my opinion, the archaeological data on which these interpretations are based, are not presented in an adequate manner. With the exception of the total number of lithics of each archaeological horizon (AH) no numeric values are specified in the main text. Even in the Supporting Online Material, except for Fig. S2 and Table S1 no further information about the stone artefact assemblages is given. Table S1: Basic characteristics of lithic assemblages from Buhais Rockshelter, only shows values for the blank production. Some tables with metrical data and amount of tool-types, platform types etc. would be helpful and are in my opinion essential. The reader has to believe the description of the assemblages. But what does it mean that “Evidence for classic Levallois reduction systems are notably scarce in AH II”? Consequently, the results are not transparent and it is also not possible to compare the assemblages with assemblages from other sites. Important results of the paper (and absolutely worth publishing) are:

- the indication of a local Middle Palaeolithic tradition during times of high aridity is an important research progress.
- evidence of human presence and inhabitable environments in SE Arabia during the last glacial period.

- indications of a LMP population replacement.

However, results and conclusions have to be verifiable and this demands a presentation of data. This has been done for the dating- and the palaeo-environmental section of the paper but is missing for the archaeological part. It should be easy to add this information (for example some tables in the SOM), but it is a requirement for publication.

Apart from that, there are only some minor questions and comments to specific text passages:

p.3, l. 3: mMarine isotope sStage (MIS) 5

p.6: "During the excavations, seven sediment layers were identified based on their composition and designated as Geological Horizons (GH) 1 to 3. GH1 was further subdivided into four sublayers (a-d), each exhibiting slightly different characteristics (Figs. 4, S1)." This is a little bit confusing. Rewording might help (During the excavations, three Geological Horizons (GH) were identified based on their composition and designated as GH 1 to 3. GH1 was further subdivided.....). Figure 4. Is the naming of the Geological Horizons correct (GH1, GH1a, GH1b & GH1c) Shouldn't it be GH1a to GH1d?

p.8, l.4: Without reading the SOM, it is hard to understand the naming of AH la/l.

p.8: "All flake and blade types were produced by hard stone hammer percussion". Why? What is the evidence for this?

p.8: "Tool typologies include typical MP types such as notches and denticulates as well as sidescrapers, and short atypical endscrapers." How are notches and denticulates defined? What are the total numbers of tool types?

p.10: "The tool assemblage comprises typical UP types such as endscrapers, burins and truncations." Numbers?

p.20: "Every quarter square was excavated to a depth of about 3 cm, which fills about one 12 l bucket." What is the smallest excavation unit? Quarter square x 3cm? One 12 l bucket? And why "about 3 cm" and "about one bucket"? Shouldn't it be "exactly 3 cm & about one bucket" or rather "about 3 cm & exactly one bucket"? Maybe, it is just my fallacy of thinking.

p.21: "Huweimi (Fig. 9) ". Fig. 9? Is this correct?

Version 1:

Reviewer comments:

Reviewer #1

(Remarks to the Author)

I am satisfied with the revision, and recommend publication.

Reviewer #2

(Remarks to the Author)

I think that the authors have addressed the review suggestions well and that the paper is ready to be published.

Reviewer #3

(Remarks to the Author)

I appreciate the careful reworking of the manuscript by the authors. My main concern had been the, in my opinion, inadequate presentation of the archaeological material. The presented data did not allow the reader to comprehend the interpretation of the material by the authors and, finally, also not their conclusion. This did not do justice to the significance of the site with is an important key-stratigraphy for the region.

The revised version fulfils my expectations in every way. For this reason, I can approve the publication of the manuscript without any objections.

With many thanks (to authors and editor) and best wishes

Response to reviewer questions and comments

Reviewer #1 (Remarks to the Author):

Comments on the manuscript “Evidence from Buhais Rockshelter reveals human settlement in Arabia between 60,000 and 16,000 years ago” by Bretzke et al.

This study reports OSL-based human occupations at ~125, 59, 35, and 16 ka at Buhais Rockshelter (Arabian Peninsula), challenging the previous Last-Glacial abandonment models and reframing Arabian habitability and human dispersal. It fits the scope of the NC, and could have wide readership. I will suggest major to moderate revision before publication.

My review centers on two pillars: (1) the robustness of the chronological framework that underpins all archaeological and palaeoenvironmental interpretations, and (2) the clarity and readability about field-wide significance and cross-disciplinary relevance expected by NC.

First, the authors should strengthen validation of the OSL ages:

1. The study’s chronology places substantial weight on statistical age models (MAM/FMM). While the authors explain key parameter settings (e.g., σ_β), these models rest on specific assumptions and empirically set inputs. Model application should be guided not only by statistical grounds but also by the depositional context of each sample and any additional pertinent information (e.g., independent age control).

Regarding D_e analysis, two major concerns need to be addressed:

Enhance geomorphological justification for age model application. For example, the authors substitute MAM with FMM for sample BUHAIS3, hypothesizing three distinct grain input processes: aeolian influx (well-bleached), rock face weathering (unzeroed), and post-depositional mixing (younger mixing). While this reasoning is plausible, the processes proposed, e.g., weathering and bioturbation leading to discrete components, require dedicated discussion and empirical support from sedimentary or geomorphological evidence.

-> We thank the reviewer for pointing this out. However, in luminescence dating the choice of different age models is generally based on statistical observations and there are dozens of articles describing and justifying this (e.g. Galbraith and Roberts 2012, Quat. Geochron.; Chamberlain et al. 2018, Rad. Meas.; Peng 2021, Geochronometria). For sample BUHAIS3 we try to explain the origin of the D_e distribution. The reviewer asks for empirical evidence proofing the hypothesis but without any suggestion how this could be done. One could possibly distinguish between grains of aeolian influx and rock face weathering by investigating these using scanning electron microscope images but very unlikely the ‘young’ grains (post-sedimentary mixing) that led to using the FMM instead of the MAM.

The absence of independent age control for cross-validation constitutes a significant weakness in the chronological assessment. The author should acknowledge this limitation or integrate independent chronological controls (if available) for validation.

-> We thank the reviewer for pointing this out, as it provides an opportunity to explain our sampling strategy in greater detail. Sampling was guided by the site's sedimentological and archaeological stratigraphy. Over three different excavation seasons, we collected samples from different parts of the excavation area. The primary aim was to sample the same archaeological layer from different parts of the excavation from multiple locations. Particular emphasis was placed on layers containing archaeological material that is currently rare in SE Arabia (e.g. UP archaeological evidence), while additional layers were sampled to complement the site's chrono-stratigraphic framework. We argue that the fact that all OSL dating results are in correct chronological order and consistent with the site's stratigraphy provides strong support for the reliability of resulting chronological interpretation. Further support for our chronological interpretation is provided by archaeological horizon III, which aligns closely, with results from Jebel Faya. In addition, comparison with records from adjacent regions indicate that the blade technology identified in the UP layers is fully consistent with the proposed age range.

-> To clarify this point, we have revised the methods section (page 23) as follows: "Key strata such as the UP-containing GHs 1 and 1a have been repeatedly sampled from different parts of the excavation to ensure validity of the results. In total we collected ten samples from Buhais Rockshelter. Sample Buhais 2 has been excluded here since it is from a stratigraphic position above the sequence presented here linked to the Holocene occupation of the site. All produced OSL results are in chronological order and correspond with the sequence of layer deposition."

2. For the D_r calculation, the authors should provide a clear and evidence-based justification for the assumed water content values (e.g., comparison with regional moisture measurements, soil properties, or saturation limits used in similar regional studies) or explain the rationale for the assumed ranges ($3\pm 3\%$ and $4\pm 4\%$) – how did you integrate climate change and geological history into this value/error? What was the field water content in comparison?

-> We agree with the reviewer that average water content during burial represents one of the largest uncertainties in OSL dating but there is no approach that would allow to determine it. We follow several previous studies (some now cited in the supplements) that used similar values, although none of those provided clear and evidence-based justification. Anyway, using different water contents within a reasonable range ($< 10\%$) will change ages by not more than 2-3 % and have no significant effect on the interpretation of the data. We have added some remarks on this to the supplemental information.

Second, while the manuscript's technical rigor is strong and detailed, the Discussion's through-line seems fragment, diluting the translation of the chronological results into

the narrative readability for capturing new noteworthy findings and perspectives. The through-line thereby requires strengthening.

-> We thank the reviewer for this constructive critique and agree that our previously chosen approach to discuss our findings could be improved to make it easier accessible for the broad readership of Nature Communication. We follow the reviewer's suggestions below and re-organized and extended the discussion part. -> see pp. 13-21.

Clarify a stepwise pathway for challenging Last-Glacial absence models: why the consensus held the interior to be unpopulated in hyper-arid periods (MIS 4/3/2), how stratigraphic scarcity and regional climate interpretations shaped that view, and how your 60–16 ka chronology reframes it via multi-pulses of terminal-Pleistocene occupation with short-lived hydrological conditions.

-> We appreciate the suggestion of an alternative approach to the discussion. We followed this recommendation and changed the entire discussion according to the reviewer's suggestions. See discussion on pages 13-21 in the revised manuscript.

Once the core findings are sharpened, the compilation of dated archaeological sites should enable revisions to dispersal routes and late-Pleistocene settlement dynamics, which could be delineated further in both text and map. For regional comparison with “northern corridor” frameworks, a similar work of OSL-based MIS 5 Levant wetland corridors by Abbas et al. (2023, Human dispersals out of Africa via the Levant, Science Advances) would be helpful.

-> We agree with reviewer 1 that visualization of conclusions would improve communication of our results. We have created a new figure (Fig. 10, page 17) to visualize the main conclusions regarding potential spatial connectivity of populations settling in SE Arabia during the last glacial period.

Make the cross-disciplinary bridge explicit by showing how the Buhais sequence serves as an archaeological anchor for evaluating the hypothesized Basal Eurasian population; temper claims where evidence is indirect.

-> We considered this suggestion and have extended the respective paragraph on pages 20-21 as follows: “Our results demonstrate that one of the last pre-50 ka populations in SW Asia can be found in SE Arabia at Buhais Rockshelter. Presented chronological and cultural records form an empirical basis that provides a first anchor point for further cross-disciplinary research into deciphering the Basal Eurasian population and human's last glacial history in SW Asia.”

A few missing references with OSL ages of Arabian water-lain sediments in the last glacial:

1. Steffen Mischke et al., 2021. A Late Pleistocene Wetland Setting in the Arid Jurf ed Darawish Region in Central Jordan. *Frontiers in Earth Sciences*. doi: 10.3389/feart.2021.722435

2. Bety S. Al-Saqarat, et al., 2020. A wetland oasis at Wadi Gharandal spanning 125–70 ka on the human migration trail in southern Jordan. *Quaternary Research*, <https://doi.org/10.1017/qua.2020.82>

3. Abbas M et al., 2023. Human dispersals out of Africa via the Levant. *Science Advances*, <https://www.science.org/doi/10.1126/sciadv.adi6838>

Thanks for these suggestions, we have added all of them.

Reviewer #2 (Remarks to the Author):

This manuscript reports new information from the Buhais Rockshelter in the United Arab Emirates. While the work reported is scientifically robust and has the potential to be significant, it comes across largely as a site report and it does not contribute a substantive, novel, argument or testing of hypotheses. For instance, take the title, ‘Evidence from Buhais Rockshelter reveals human settlement in Arabia between 60,000 and 16,000 years ago’. Given that several sites are already known in this time period in Arabia (e.g. SD-1, Matafah), what is fundamentally novel about the new findings? The most important part of the study is perhaps the discovery of Upper Paleolithic material – yet the authors say they are working on a separate paper about this (401), so not much information is given.

-> We thank reviewer 2 for pointing this out, which gives us the opportunity for clarification. The intention of this paper is the presentation of new findings from SE Arabia, which we believe have multiple important implications that are of interest for a wider readership. We do not agree that three sites in Arabia (2 for the 60-ka window, 1 for the 35-ka and none for the 16-ka window) can be excepted as being representative for 44,000 years of history in an area of about 3 Mio km². The two sites for the 60-ka window for example are located about 1,400 km apart. While the southern one (SD-1) indeed provides substantial data (21m² excavated area, 9 OSL dates, 5488 lithic artifacts), the northern one (ALM 3) provides much less extensive information (ca. 2m² excavated area, 2 OSL dates, 103 lithic artifacts from stratified context). The one site mentioned for the 35-ka window (Matafah) is a 3 m² test excavation with 89 lithics where the original excavator claims that they have too few finds to characterize the lithic tradition. Moreover, all these sites represent occupation events from just one time slice. In contrast, our record now provides diachronic information from one locality for the first time. This is currently exceptional in Arabia considering the period 60-12 ka and allows conclusions about local continuity or discontinuity in the settlement of a region, which also have inter-regional implications when compared with available records.

We added information about the extent of excavation and finds from Matafah and extended the discussion to make it clearer why our data provides substantially new insights into Arabian last glacial history. -> see p. 19 and discussion pp. 13-21.

Regarding the comment on the UP material, we removed this phrase and added more data to provide a more comprehensive insight into the recovered UP material. This is in line with a suggestion by reviewer 3. Please see SI.

Care is also needed with strawman arguments, namely they claim to “contradict the prevailing view of human absence in Arabia at the end of the Pleistocene”. The reference given for this is a paper about a speleothem record from Yemen. That paper argues wet periods facilitated dispersals into Arabia, but it is a strawman argument to claim that this study in particular, and the ‘general view’ more broadly, is that there were no humans in Arabia at the end of the Pleistocene. The work of Rose, Hilbert etc has long argued for a terminal Pleistocene human presence in southeast Arabia.

-> We agree with the reviewer’s point that there are previous arguments for terminal Pleistocene occupation of southern Arabia. These, however, consider, to our knowledge, mainly the latest part of the Pleistocene (~12 ka). We have re-written this part and added more references that argue for scarce if not lack of occupation. We write now on page 14:

“This in addition to scarce archaeological records from this period, often leads to the assumption of a largely uninhabitable Arabian Peninsula between approximately 60 ka and 16 ka^{2,27,28}.”

So while the study reports important findings, it does not seem like these are presented in a particular useful way, it is more like a site report with some vague indications of how the findings relate to things like adaptation to arid conditions and ‘distinctive patterns of settlement dynamics’ (page 3). Fundamentally, the story is seemingly one of repeated migrations into the area in association with wet periods. The authors argue against Late Pleistocene population continuity in Arabia. Instead, it looks like they have identified pulses of hominin dispersal into Arabia, largely correlating with those already known (e.g. SD-1 ca 60 ka, Matafah ca 33 ka). They are also present a somewhat contradictory argument by attacking the point that dispersals into Arabia correlate with wet phases, by saying their studies show the evidence of people in southern Arabia “during the presumably arid last glacial period”, but directly contradict themselves by finding that actually the periods of occupation correlate with “repeated reactivation of freshwater springs and hydrological systems”. What their findings suggest is that the European-derived glacial-interglacial dichotomy may not be the most useful framework for low latitude palaeohydrological change – as indeed many papers have argued in recent years. They themselves point out how actually the regional sequence of SE Arabia suggests human presence in association with Monsoon Index Peaks, when rainfall increased in the area.

-> We thank the reviewer for considering our results important. To further strengthen our communication, we re-organized and extended the discussion to make implications of our findings better understandable. Please see discussion pp. 13-21.

Regarding potential contradictory claims about occupation in SE Arabia during the supposedly arid phase of the last glacial period, we agree with the reviewer that this claim can be misunderstandable and is based on a simplified view. To take account of the obvious complexity of the problem, we added available information from both climate models and local records to explain the background in greater detail. We removed the phrase “supposedly hyper-arid conditions” -> see abstract, p. 1 and discussion pp.13-14.

The ‘nuts and bolts’ of the paper are good, such as the figures and the general descriptions of the deposits, their associated archaeological assemblages, and luminescence age estimates are all clearly presented and logical. The problem is the lack of orientation of the study to the key research themes in the region.

-> We are pleased to note that the reviewer appreciates our goal of basing the presentation of our results on solid archaeological, chronological and palaeoenvironmental field work data. We used all reviewer comments to work extensively on the presentation of the implications of our findings and re-organized and extended the discussion. We believe that the new discussion now provides a better orientation of our study to key research themes in Arabia -> please see the discussion part pp. 13-21.

Reviewer #3 (Remarks to the Author):

The role of the Arabian Peninsula in the expansion of anatomically modern humans is a hotly debated topic in archaeological and palaeo-anthropological research. The “Southern dispersal route” is suggested by genetic evidence, but sound archaeological evidence is missing up to now. Therefore, any data from this region and this period of time is most welcome. Especially the absence of stratified deposits, covering the transition from Middle Palaeolithic to Upper Palaeolithic stone artefact traditions during the Late Pleistocene made any conclusions about this important step in human history impossible. Therefore, Buhais Rockshelter is a key-site with supra-regional relevance. The presented results are of utmost interest for the scientific community and an appropriate publication absolutely desirable. This leads me to my main point of criticism. In my opinion, the archaeological data on which these interpretations are based, are not presented in an adequate manner. With the exception of the total number of lithics of each archaeological horizon (AH) no numeric values are specified in the main text. Even in the Supporting Online Material, except for Fig. S2 and Table S1 no further information about the stone artefact assemblages is given. Table S1: Basic characteristics of lithic assemblages from Buhais Rockshelter, only shows values for the

blank production. Some tables with metrical data and amount of tool-types, platform types etc. would be helpful and are in my opinion essential. The reader has to believe the description of the assemblages. But what does it mean that “Evidence for classic Levallois reduction systems are notably scarce in AH II”? Consequently, the results are not transparent and it is also not possible to compare the assemblages with assemblages from other sites.

-> We appreciate that reviewer 3 raised this point. This gives us the opportunity to add more detailed archaeological data. We fully agree that the requested data is of importance for understanding the different material cultures at the site and are more than happy to add more specialized data for the expert part of the readership of Nature Communications.

Important results of the paper (and absolutely worth publishing) are:

- the indication of a local Middle Palaeolithic tradition during times of high aridity is an important research progress.
- evidence of human presence and inhabitable environments in SE Arabia during the last glacial period.
- indications of a LMP population replacement.

However, results and conclusions have to be verifiable and this demands a presentation of data. This has been done for the dating- and the palaeo-environmental section of the paper but is missing for the archaeological part. It should be easy to add this information (for example some tables in the SOM), but it is a requirement for publication.

-> We follow this recommendation and have added figures 3-7 and table 2 to the Supplementary Information.

Apart from that, there are only some minor questions and comments to specific text passages:

p.3, l. 3: mMarine isotope sStage (MIS) 5

-> done

p.6: “During the excavations, seven sediment layers were identified based on their composition and designated as Geological Horizons (GH) 1 to 3. GH1 was further subdivided into four sublayers (a-d), each exhibiting slightly different characteristics (Figs. 4, S1).” This is a little bit confusing. Rewording might help (During the excavations, three Geological Horizons (GH) were identified based on their composition and designated as GH 1 to 3. GH1 was further subdivided.....).

-> done

Figure 4. Is the naming of the Geological Horizons correct (GH1, GH1a, GH1b & GH1c) Shouldn't it be GH1a to GH1d?

-> no, since GH 1d is just a local lens visible exclusively in the western profile, please see *Supplementary Fig. 1*.

p.8, l.4: Without reading the SOM, it is hard to understand the naming of AH Ia/I.

-> thank you for pointing this out, we added a note after the first occurrence of AH Ia to make the reader aware of additional information, we added "(see *Supplementary Site Information for details on the nomenclature*)" after the first mentioning of the term AH Ia on page 7.

p.8: "All flake and blade types were produced by hard stone hammer percussion". Why? What is the evidence for this?

-> Hard stone hammer percussion is typically identified from morphological features of the flakes and blades such as pronounced bulbs of percussion and relatively thick striking platforms. We have added this information to the text on page 7.

p.8: "Tool typologies include typical MP types such as notches and denticulates as well as sidescrapers, and short atypical endscrapers." How are notches and denticulates defined? What are the total numbers of tool types?

-> We follow here the standard definition as for example provided by the "*Handbook of Paleolithic Typology*" (Debénath and Dibble, 1994). Notches are lithic artifacts with one notch near the edge of the artifact while denticulates feature two and more notches often creating toothed edges. We have added numerical information to the supplementary information.

p.10: "The tool assemblage comprises typical UP types such as endscrapers, burins and truncations." Numbers?

-> We have added information on the typology to the supplementary information, which now also includes more numerical information. Please see *Supplementary table 2 and Supplementary fig. 3*.

p.20: "Every quarter square was excavated to a depth of about 3 cm, which fills about one 12 l bucket." What is the smallest excavation unit? Quarter square x 3cm? One 12 l bucket? And why "about 3 cm" and "about one bucket"? Shouldn't it be "exactly 3 cm & about one bucket" or rather "about 3 cm & exactly one bucket"? Maybe, it is just my fallacy of thinking.

-> the smallest unit is 25cm x 25cm x one 12l bucket, the 3cm are mentioned to give an impression of the progress. Very often it is about 3 cm. The normal approach is that we aim at filling one 12l bucket in one round, but it is never exact, since we must take multiple aspects into account when digging. First we follow the natural slope and thickness of the sediment (and we stop in case we reach a new layer), second we try to reduce the entire excavated area as one, which means there should be no steps to the neighboring $\frac{1}{4}$ squares. Third the amount of sediment excavated in one round is also related to the size of rock fragments that are part of the excavated sediments. Taken all this together will in reality not allow exact 12l to be excavated all the time. This is why we used the wording as it is.

p.21: "Huweimi (Fig. 9)". Fig. 9? Is this correct?

-> thanks for spotting this, we changed the reference to figure 8. Now on page 22.